# Humans adapt their anticipatory eye movements to the volatility of visual motion properties

**Chloé Pasturel, Anna Montagnini [ID], Laurent Udo Perrinet [ID]** *

Institut de Neurosciences de la Timone (UMR 7289), Aix Marseille Univ, CNRS, Marseille, France

* Laurent.Perrinet@univ-amu.fr

**Data Availability Statement:** All raw experimental data and codes used for data analysis and model simulations are freely available at: https://github.com/laurentperrinet/PasturelMontagniniPerrinet2020.

## Abstract

Animal behavior constantly adapts to changes, for example when the statistical properties of the environment change unexpectedly. For an agent that interacts with this volatile setting, it is important to react accurately and as quickly as possible. It has already been shown that when a random sequence of motion ramps of a visual target is biased to one direction (e.g. right or left), human observers adapt their eye movements to accurately anticipate the target's expected direction. Here, we prove that this ability extends to a volatile environment where the probability bias could change at random switching times. In addition, we also recorded the explicit prediction of the next outcome as reported by observers using a rating scale. Both results were compared to the estimates of a probabilistic agent that is optimal in relation to the assumed generative model. Compared to the classical leaky integrator model, we found a better match between our probabilistic agent and the behavioral responses, both for the anticipatory eye movements and the explicit task. Furthermore, by controlling the level of preference between exploitation and exploration in the model, we were able to fit for each individual's experimental dataset the most likely level of volatility and analyze inter-individual variability across participants. These results prove that in such an unstable environment, human observers can still represent an internal belief about the environmental contingencies, and use this representation both for sensory-motor control and for explicit judgments. This work offers an innovative approach to more generically test the diversity of human cognitive abilities in uncertain and dynamic environments.

## Author summary

Understanding how humans adapt to changing environments to make judgments or plan motor responses based on time-varying sensory information is crucial for psychology, neuroscience and artificial intelligence. Current theories for how we deal with the environment's uncertainty, that is, in response to the introduction of some randomness change, mostly rely on the behavior at equilibrium, long after after a change. Here, we show that in the more ecological case where the context switches at random times all along the experiment, an adaptation to this volatility can be performed online. In

**Funding:** This work was supported by EU Marie-Sklodowska-Curie Grant No 642961 (PACE-ITN / A. Montagnini and L. Perrinet as participants) and by the Fondation pour le Recherche Médicale, under the program Équipe FRM (DEQ20180339203/ PredictEye/PI: G Masson/ A. Montagnini and L. Perrinet as participants). The funders had no role in study design, data collection and analysis, decision to publish, or preparation of the manuscript.

**Competing interests:** The authors have declared that no competing interests exist.

particular, we show in two behavioral experiments that humans can adapt to such volatility at the early sensorimotor level, through their anticipatory eye movements, but also at a higher cognitive level, through explicit ratings. Our results suggest that humans (and future artificial systems) can use much richer adaptive strategies than previously assumed.

## Introduction

### Volatility of sensory contingencies and the adaptation of cognitive systems

We live in a fundamentally volatile world for which our cognitive system has to constantly adapt. In particular, this volatility may be generated by processes with different time scales. Imagine for instance you are a general practitioner and that you usually report an average number of three persons infected by flu per week. However, this rate is variable and over the past week you observe that the rate increased to ten cases. In consequence, two alternative interpretations are available: the first possibility is that there is an outbreak of flu and one should then estimate its incidence (i.e. the rate of new cases) since the inferred outbreak's onset, in order to quantify the infection rate specific to this outbreak, but also to update the value of the probability of a new outbreak at a longer time scale. Alternatively, these cases are "unlucky" coincidences that originate from the natural variability of the underlying statistical process which drive patients to the doctor, and which are instances drawn from the same stationary random process. In that option, it may be possible to readjust the estimated baseline rate of infection with this new data. This example illustrates one fundamental problem with which our cognitive system is faced: when observing new sensory evidence, *should I stay* and continue to exploit this novel data with respect to my current beliefs about the environment's state or *should I go* and explore a new hypothesis about the random process generating the observations since the detection of a switch in the environment?

This uncertainty in the environment's state is characterized by its *volatility* which by definition measures the temporal variability of the sufficient parameters of a random variable. Such meta-analysis of the environment's statistical properties is an effective strategy at a large temporal scale level, as that for the flu outbreak of our example, but also at all levels which are behaviorally relevant, such as contextual changes in our everyday life. Inferring near-future states in a dynamic environment, such that one can prepare to act upon them ahead of their occurrence—or at least forming beliefs as precise as possible about a future environmental context—is an ubiquitous challenge for cognitive systems [1]. In the long term, how the human brain dynamically manages this trade-off between exploitation and exploration is essential to the adaptation of the behavior through reinforcement learning [2].

In controlled psychophysical experimental settings which challenge visual perception or sensorimotor associations, such adaptive processes have been mostly put in evidence by precisely analyzing the participants' behavior in a sequence of experimental trials. These typically highlight sequential effects at the time scale of several seconds to minutes or even hours in the case of the adaptation to a persistent sensorimotor relation. Indeed, stimulus history of sensory events influences how the current stimulus is perceived [3–7] and acted upon [8–11]. Two qualitatively opposite effects of the stimulus history have been described: negative (adaptation), and positive (priming-like) effects. Adaptation reduces the sensitivity to recurrently presented stimuli, thus yielding a re-calibrated perceptual experience [12–14]. On the other hand, priming is a facilitatory effect that enhances the identification of repeated stimuli [15, 16]: in sensorimotor control, the same stimulus presented several times could indeed lead to faster and more precise responses. Interestingly, priming effects are sometimes paralleled by

anticipatory motor responses which are positively correlated with the repetition of stimulus properties. A well-known example of this behavior are anticipatory smooth eye movements (aSPEM or shortly, anticipatory pursuit), as we will illustrate in the next section.

Overall, the ability to take into account statistical regularities in the event sequence appears as a fundamental ability for the adaptive behavior of living species. Importantly, few studies have addressed the question of whether the estimate of such regularities is explicit, and whether such explicit reports of the dynamic statistical estimates would eventually correlate with the measures of behavioral adaptation or priming. Here we aim at investigating this question in the specific case of the processing of a target's motion direction. In addition, we attempt to palliate the lack of a solid modeling approach to best understand the computation underlying behavioral adaptation to the environment's statistics, and in particular how sequential effects are integrated within a hierarchical statistical framework.

Bayesian inference offers an effective methodology to deal with this question. Indeed, these methods allow to define and quantitatively assess a range of hypotheses about the processing of possibly noisy information by some formal agents [17–19]. A key principle in the Bayesian inference approach is to introduce so-called *latent variables* which explicitly represent different hypotheses by the agent and how these may predict experimental outcomes. Each hypothesis defines different weights in the graph of probabilistic dependencies between variables (for instance between the number of patients at a practitioner and the reality of a flu pandemic). Then, using the rules of probability calculus and knowing incoming measurements, one can progressively update beliefs about the latent variables, and eventually infer the hidden structure underlying the received inputs [20, 21]. For instance, using Bayes's rule, one can combine the likelihood of observations given a given generative model and the prior on these latent variables [22] such that beliefs about latent variables may be represented as probabilities. Of particular interest for us is the possibility to quantitatively represent in this kind of probabilistic model the predictive and iterative nature of a sequence of events. Indeed, once the belief about latent variables is formed from the sensory input, this belief can be used to update the prior over future beliefs [23]. In such models, the comparison between expectations and actual data leads to continuous updates of the estimates of the latent variables, but also of the validity of the model. There are numerous examples of Bayesian approaches applied to the study of the adaptation to volatility. For instance, Meyniel et al [24] simulated a hierarchical Bayesian model over five previously published datasets [25–29] in the domain of cognitive neuroscience. Here we focus on an extension of this approach to the study of motion processing and eye movements.

## Anticipatory Smooth Pursuit Eye Movements (aSPEM)

Humans are able to accurately track a moving object with a combination of saccades and Smooth Pursuit Eye Movements (for a review see [30]). These movements allow us to align and stabilize the object on the fovea, thus enabling high-resolution visual processing. This process is delayed by different factors such as axonal transduction, neural processing latencies and the inertia of the oculomotor system [31]. When predictive information is available about target's motion, an anticipatory Smooth Pursuit Eye Movement (aSPEM or shortly, anticipatory pursuit) is generated before its appearance [32–34] thereby reducing visuomotor latency [35]. Moreover, some experiments have demonstrated the existence of prediction-based smooth pursuit maintenance during the transient disappearance of a moving target [36–38] and even predictive acceleration or deceleration during visual tracking [37, 39]. Overall, although the initiation of smooth pursuit eye movements is almost always driven by a visual motion signal, it is now clear that smooth pursuit behavior can be modulated at different stages by extra-

retinal, predictive information even in the absence of a direct visual stimulation [40]. Several functional and computational models have been proposed in the literature for the different forms of prediction-based smooth eye movements, such as zero-lag tracking of a periodic target [41] or pursuit maintenance during target occlusion [39]. More recently an effort has been made to provide a more general theoretical framework, which is based on Bayesian inference and the reliability-based cue combination. Although the mapping of this theoretical framework onto neuronal functions remains to be elucidated, it has the clear advantage of generality, as for instance, it would encompass all forms of smooth pursuit behavior, including prediction-based and visually-guided tracking [42–45]. Here, we present a model extending this recent theoretical effort to include the adaptivity to a volatile environment.

Experience-based anticipatory pursuit behavior is remarkable in different aspects. First, its buildup is relatively fast, such that only a few trials are sufficient to observe the effects of specific regularities in the properties of visual motion, such as speed, timing or direction [10, 44, 46]. Second, it is a robust phenomenon, which has been observed on a large population of human participants and even in non-human primates (for a recent review see [47]). Note also, that human participants seem to be largely unaware of this behavior (as inferred from informal questioning). Finally, this kind of behavior has proven to be exquisitely sensitive to the probabilistic properties of the sensorimotor context.

Typically, anticipatory pursuit is observed after a temporal cue and before target motion onset [33, 34, 46]. In previous studies [11, 48], we have analyzed how forthcoming motion properties, such as target speed or direction, can be anticipated with coherently oriented eye movements. We have observed that the amplitude of anticipation, as measured by the mean anticipatory eye velocity, increases when the target repeatedly moves in the same direction. In particular, the mean anticipatory eye velocity is linearly related to the probability of motion's speed or direction. These results are coherent with findings by other groups [46, 49–51] and they indicate that anticipatory pursuit behavior is potentially a useful marker to study the internal representation of motion expectancy, and in particular to analyze how such expectancy is dynamically modulated by probabilistic contingencies in shaping oculomotor behavior.

## Contributions

The goal of this study is to generalize the adaptive process observed in anticipatory pursuit [48, 51] to more ecological settings and also to broaden its scope by showing that such adaptive processes occur also at an explicit level. We already mentioned that by manipulating the probability bias for target motion direction, it is possible to modulate the strength (direction and mean velocity) of anticipatory pursuit. This suggests that probabilistic information about direction bias may be used to inform the internal representation of motion prediction for the initiation of anticipatory movements. However, previous studies have overlooked the importance to design a realistic generative model to dynamically manipulate the probability bias and generate an ecologically relevant input sequence of target directions. A possible confound comes from the fact that previous studies have used fixed-lengths sequences of trials, stacked in a sequence of conditions defined by the different probability biases. Indeed, observers may potentially pick up the information on the block's length to predict the occurrence of a switch (a change in probability bias) during the experiment. Second, we observed qualitatively that following a switch, the amplitude of anticipatory pursuit velocity changed gradually, consistently with other adaptation paradigms [52–54]. The estimate of the characteristic temporal parameters for this adaptation mechanism may become particularly challenging in a dynamic context, where the probabilistic contingencies vary in time in an unpredictable way. Finally,

whether and how the information processing underlying the buildup of anticipatory pursuit and its dynamics is linked to an explicit estimate of probabilities is still largely unknown.

To assess the dynamics of the adaptive processes which compensate for the variability within sensory sequences, one may generate random sequences of Target Directions (TDs) using a dynamic value for the probability bias $p = Pr(TD$ is 'right'), with a parametric mechanism controlling for the volatility at each trial. In the Hierarchical Gaussian Filter model [55], for instance, volatility is controlled as a non-linear transformation of a random walk (modeled itself by a Brownian motion with a given diffusion coefficient). Ultimately, this hierarchical model allows to generate a sequence of binary choices where volatility is controlled by a specific random variable which fluctuates in time according to some probabilistic law. Such a forward probabilistic model is invertible using some simplifying assumptions and allows to extract a time-varying inference of the agent's belief about volatility [56]. Herein, to explicitly analyze the effect of history length, we rather extend the protocol of [48] such that the probability bias is still fixed within sub-blocks but that these sub-blocks have variable lengths, that is, by introducing switches occurring at random times. Therefore, similarly to [57], we use a model for which the bias $p$ in target direction varies according to a piecewise-constant function. We expect that within each of these sub-blocks that we defined, the uncertainty about of the value of $p$ will progressively decrease as we accumulate trials. In addition, the range of possible biases was finite ($p \in \{0, .1, .25, .5, .75, .9, 1\}$) in our previous study. In the present work, we also extend the paradigm by drawing $p$ as a continuous random variable within the whole range of possible probability biases (that is, the segment [0, 1]).

As a summary, we first draw random events (that we denote as "switches") with a given mean frequency (the "hazard rate") and which controls the strength of the volatility. Second, the value $p$ of the bias only changes at the moment of a switch, independently of the previous bias' value and it is stationary between two switches, forming what we call an "epoch". Third, target direction is drawn as a Bernoulli trial using the current value of $p$. Such a hierarchical structure is presented in Fig 1A, where we show the realization of the target's directions

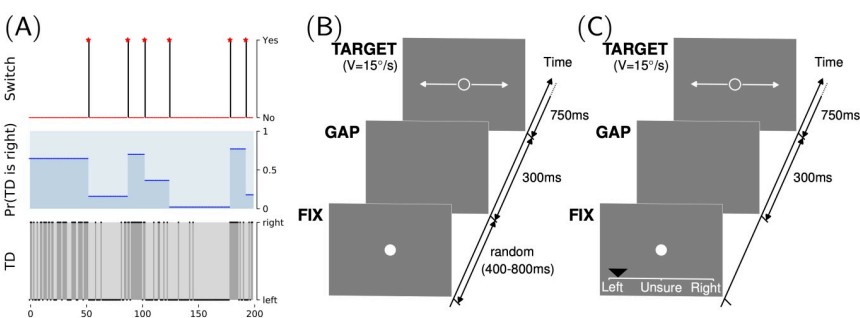

**Fig 1. Smooth pursuit eye movements and explicit direction predictions in a volatile switching environment.** We test the capacity of human participants to adapt to a volatile environment. *(A)* We use a 3-layered generative model of fluctuations in target directions (TD) that we call the Binary Switching model. This TD binary variable is chosen using a Bernoulli trial of a given probability bias. This probability bias is constant for as many trials until a switch is generated. At a switch, the bias is chosen at random from a given prior. Switches are generated in the third layer as binary events drawn from a Bernoulli trial with a given hazard rate (defined here as 1/40 per trial). We show one realization of a block of 200 trials. *(B)* The eye-movements task was an adapted version of a task developed by [48]. Each trial consisted of sequentially: a fixation dot (FIX, of random duration between 400 and 800 ms), a blank screen (GAP, of fixed duration of 300 ms) and a moving ring-shaped target (TARGET, with 15°/s velocity) which the observers were instructed to follow. The direction of the target (right or left) was drawn pseudo-randomly according to the generative model defined above. *(C)* In order to titrate the adaptation to the environmental volatility of target direction at the explicit and conscious level, we invited each observer to perform on a different day a new variant of the direction-biased experiment, where we asked participants to predict, *before each trial*, their estimate of the forthcoming direction of the target. As shown in this sample screenshot, this was performed by moving a mouse cursor (black triangle) on a continuous rating scale between "Left", to "Unsure" and finally "Right".

sequence, the trajectory of the underlying probability bias (hidden to the observer), and the occurrences of switches. Mathematically, this can be considered as a three-layered hierarchical model defining the evolution of the model at each trial $t$ as the vector $(x_2^t, x_1^t, x_0^t)$. At the topmost layer, the occurrence $x_2^t \in \{0, 1\}$ of a switch (1 for true, 0 for false) is drawn from a Bernoulli trial $\mathcal{B}$ parameterized by its hazard rate $h$ (as the frequency of occurrence for each trial). The value of $\tau = \frac{1}{h}$ thus gives the average duration (in number of trials) between the occurrence of two switches. In the middle layer, the probability bias $p$ of target direction is a random variable that we define as $x_1^t \in [0, 1]$. It is chosen at random from a prior distribution $\mathcal{P}$ at the moment of a switch, and else it is constant until the next occurrence of a switch. The prior distribution $\mathcal{P}$ can be for instance the uniform distribution $\mathcal{U}$ on $[0, 1]$ or Jeffrey's prior $\mathcal{P}$ (see Appendix). Finally, a target moves either to the left or to the right, and we denote this variable (target direction, TD) as $x_0^t \in \{0, 1\}$. This direction is drawn from a Bernoulli trial parameterized by the direction bias $p = x_1^t$. In short, this is described according to the following equations:

$$
\begin{cases}
\text{Occurrence of a switch}: x_2^t \propto \mathcal{B}(1/\tau) \\[1em]
\text{Dynamics of probability bias } p = x_1^t \begin{cases} \text{if} \quad x_2^t = 0 \quad \text{then} \quad x_1^t = x_1^{t-1} \\[1em] \text{else} \quad x_1^t \propto \mathcal{P} \end{cases} \\[1em]
\text{Sequence of directions}: x_0^t \propto \mathcal{B}(x_1^t)
\end{cases} \tag{1}
$$

In this study, we generated a sequence of 600 trials, and there is by construction a switch at $t = 0$ (that is, $x_2^0 = 1$). In addition, we imposed in our sequence that a switch occurs after trial numbers 200 and 400, in order to be able to compare adaptation properties across these three different trial blocks. With such a three-layered structure, the model generates the randomized occurrence of switches, itself generating epochs with constant direction probability and finally the random sequence of Target Direction (TD) occurrences at each trial. This system of three equations defined in Eq 1 defines the Binary Switching model which we used for the generation of experimental sequences presented to human participants in the experiments. We will use that generative model as the basis for an ideal observer model equipped to invert that generative model in order to estimate the time-varying probability bias for a given sequence of observations (TDs). The comparison of human behavior with the ideal observer model's predictions will allow us to test it as a model for the adaptation of human behavior to the environment's volatility.

This paper is organized in five parts. After this introduction where we presented the motivation for this study, the next section will present an inversion of the (forward) binary switching generative model, coined the Binary Bayesian Change-Point (BBCP) model. To our knowledge, such algorithm was not yet available, and we will here provide with an exact analytical solution by extending previous results from [58] to the binary nature of data in the Binary Switching model presented above (see Eq 1). In addition, the proposed algorithm is biologically realistic as it uses simple computations and is *online*, that is, all computations on the sequence may be done using solely a set of variables available at the present trial, compactly representing all the sequence history seen in previous trials. We will also provide a computational implementation and a quantitative evaluation of this algorithm. Then, we will present the analysis of experimental evidence to validate the generalization of previous results with this novel switching protocol. In order to understand the nature of the representation of motion regularities underlying adaptive behavior, we collected both the recording of eye movements and the verbal explicit judgments about expectations on motion direction. In one session,

participants were asked to estimate "how much they are confident that the target will move to the right or left in the next trial" and to adjust the cursor's position on the screen accordingly (see Fig 1C). In the other experimental session on a different day, we showed the same sequence of target directions and recorded participants' eye movements (see Fig 1B). Combining these theoretical and experimental results, a novelty of our approach is to use the BBCP agent as a regressor which will allow us to match experimental results and to compare its predictive power compared to classical models such as the leaky integrator model. Hence, we will show that behavioral results match best with the BBCP model. In the following section, we will synthesize these results by inferring the volatility parameters inherent to the models by best-fitting it to each each individual participant. This will allow the analysis of inter-individual behavioral responses for each session. In particular, we will test if one could extract observers' prior (preferred) volatility, that is, a measure of the dynamic compromise between exploitation ("should I stay?") and exploration ("should I go?") for the two different sessions challenging predictive adaptive processes both at the implicit and explicit levels. Finally, we will summarize and conclude this study and offer some perspectives for future work.

## Results

### Binary Bayesian Change-Point (BBCP) detection model

As we saw above, Bayesian methods provide a powerful framework for studying human behavior and adaptive processes in particular. For instance, [55] first defined a multi-layered generative model for sequences of input stimuli. By inverting this stochastic forward process, they could extract relevant descriptors at the different levels of the model and fit these parameters with the recorded behavior. Here, we use a similar approach, focusing specifically on the binary switching generative model, as defined in Eq 1. To begin, we define as a control a first ideal observer, the *leaky integrator* (or *forgetful agent*), which has an exponentially-decaying memory for the events that occurred in the past trials. This agent can equivalently be described as one which assumes that volatility is stationary with a fixed characteristic frequency of switches. Then, we extend this model to an agent which assumes the existence of (randomly occurring) switches, that is, that the agent is equipped with the prior knowledge that the value of the probability bias may change at specific (yet randomly drawn) trials, as defined by the forward probabilistic model in Eq 1.

**Forgetful agent (Leaky integrator) detection model.** The leaky integrator ideal observer represents a classical, widespread and realistic model of how trial-history shapes adaptive processes in human behavior [59]. It is also well adapted to model motion expectation in the direction-biased experiment which leads to anticipatory pursuit. In this model, given the sequence of observations $x_0^t$ from trial 0 to $t$, the expectation $p = \hat{x}_1^{t+1}$ of the probability for the next trial direction can be modeled by making a simple heuristic [59]: This probability is the weighted average of the previously predicted probability, $\hat{x}_1^t$, with the new information $x_0^t$, where the weight corresponds to a leak term (or discount) equal to $(1 - h)$, with $h \in [0, 1]$. At trial $t$, this model can be expressed with the following equation:

$$\hat{x}_1^{t+1} = (1 - h) \cdot \hat{x}_1^t + h \cdot x_0^t \tag{2}$$

where $\hat{x}_1^{t=0}$ is equal to some prior value (0.5 in the unbiased case), corresponding to the best guess at $t = 0$ (prior to the observation of any data).

In other words, the predicted probability $\hat{x}_1^{t+1}$ is computed from the integration of previous instances with a progressive discount of past information. The value of the scalar $h$ represents a compromise between responding rapidly to changes in the environment ($h \approx 1$) and not prematurely discarding information still of value for slowly changing contexts ($h \approx 0$). For that

 

reason, we call this scalar the hazard rate in the same way to that defined for the binary switching generative model presented above (see Eq 1). Moreover, one can define $\tau = 1/h$ as a characteristic time (in units of number of trials) for the temporal integration of information. Looking more closely at this expression, the "forgetful agent" computed in Eq 2 consists of an exponentially-weighted moving average (see Appendix). It may thus be equivalently written in the form of a time-weighted average:

$$\hat{x}_1^{t+1} = (1-h)^{t+1} \cdot \hat{x}_1^{t=0} + h \cdot \sum_{0 \leq i \leq t} (1-h)^i \cdot x_0^{t-i} \tag{3}$$

The first term corresponds to the discounted effect of the prior value, which tends to 0 as $t$ increases. More importantly, as $1 - h < 1$, the second term corresponds to the *leaky* integration of novel observations. Inversely, let us now assume that the true probability bias for direction changes randomly with a mean rate of once every $\tau$ trials: $Pr(x_2^t = 1) = h$. As a consequence, the probability that the bias does not change is $Pr(x_2^t = 0) = 1 - h$ at each trial. Assuming independence of these occurrences, the predicted probability $p = \hat{x}_1^{t+1}$ is thus proportional to the sum of the past observations weighted by the belief that the bias has not changed during $i$ trials in the past, that is, exactly as defined by the second term of the right-hand side in Eq 3. This shows that assuming that changes occur at a constant rate ($\hat{x}_2^t = h$) but ignoring more precise information on the temporal occurrence of the switch, the optimal solution to this inference problem is the ideal observer defined in Eq 3, which finds an online recursive solution in Eq 2. We therefore proved here that the heuristic derived for the leaky integrator is an exact inversion of the two-layered generative model which assumes a constant epoch-duration between switches of the probability bias.

The correspondence that we proved between the weighted moving average heuristic and the forgetful agent model as an ideal solution to that generative model leads us to several interim conclusions. First, the time series of inferred $\hat{x}_1^{t+1}$ values can serve as a regressor for behavioral data to test whether human observers follow a similar strategy. In particular, the free parameter of the model ($h$), may be fitted to the behavioral dataset. Testing different hypothesis for the value of $h$ thus allows to infer the agents' most likely belief in the (fixed) weight decay. Now, since we have defined a first generative model and the corresponding ideal observer (the forgetful agent), we next define a more complex model, in order to overcome some of the limits of the leaky integrator. Indeed, a first criticism could be that this model is too rigid and does not sufficiently account for the dynamics of contextual changes [60] as the weight decay corresponds to assuming *a priori* a constant precision in the data sequence, contrary to more elaborate Bayesian models [61]. It seems plausible that the memory size (or history length) used by the brain to infer any event probability can vary, and that this variation could be related to an estimate of environmental volatility as inferred from past data. The model presented in Eq 3 uses a constant weight for all trials, while the actual precision of each trial can be potentially evaluated and used for precision-weighted estimation of the probability bias. To address this hypothesis, our next model is inspired by the Bayesian Change-Point detection model [58] of an ideal agent inferring the trajectory in time of the probability bias ($x_1^t$), but also predicting the probability $Pr(x_2^t = 1)$ of the occurrence of switches.

**Binary Bayesian Change-Point (BBCP) detection model.** There is a crucial difference between the forgetful agent presented above and an ideal agent which would invert the (generative) Binary Switching model (see Eq 1). Indeed, at any trial during the experiment, the agent may infer beliefs about the probability of the volatility $x_2^t$ which itself is driving the trajectory of the probability bias $x_1^t$. Knowing that the latter is piece-wise constant, an agent may have a belief over the number of trials since the last switch. This number, that is called the *run-length*

$r^t$ [58], is useful in two manners. First, it allows the agent to restrict the prediction $\hat{x}_1^{t+1}$ of $x_1^{t+1}$ only based on those samples produced since the last switch, from $t - r^t$ until $t$. Indeed, the samples $x_0^t$ which occurred before the last switch were drawn independently from the present true value $x_1^t$ and thus cannot help estimating the latter. As a consequence, the run-length is a latent variable that captures at any given trial all the hypotheses that may be occurring. Second, it is known that for this estimate, the precision (that is, the inverse of variance) on the estimate $\hat{x}_1^{t+1}$ grows linearly with the number of samples: The longer the run-length, the sharper the corresponding (probabilistic) belief. We have designed an agent inverting the binary switching generative model by extending the Bayesian Change-Point (BCP) detection model [58]. The latter model defines the agent as an inversion of a switching generative model for which the observed data (input) is Gaussian. We present here an exact solution for the case of the Binary Switching model, that is, for which the input is binary (here, left or right).

In order to define in all generality the change-point (switch) detection model, we will initially describe the fundamental steps leading to its construction, while providing the full algorithmic details in Appendix. The goal of predictive processing at trial $t$ is to infer the probability $Pr(x_0^{t+1}|x_0^{0:t})$ of the next datum knowing what has been observed until that trial (that we denote by $x_0^{0:t} = \{x_0^0, \ldots, x_0^t\}$). This prediction uses the agent's prior knowledge that data is the output of a given (stochastic) generative model (here, the Binary Switching model). To derive a Bayesian predictive model, we introduce the run-length as a latent variable which gives to the agent the possibility to represent different hypotheses about the input. We therefore draw a computational graph (see Fig 2A) where, at any trial, an hypothesis is formed on as many "nodes" than there are run-lengths. Note that run-lengths may be limited by the total number of trials $t$. As a readout, we can use this knowledge of the predictive probability conditioned on the run-length, such that one can compute the marginal predictive distribution:

$$Pr(x_0^{t+1}|x_0^{0:t}) = \sum_{r^t \geq 0} Pr(x_0^{t+1}|r^t, x_0^{0:t}) \cdot \beta_t^{(r)} \qquad (4)$$

where $Pr(x_0^{t+1}|r^t, x_0^{0:t})$ is the probability of the Bernoulli trial modeling the outcome of a future datum $x_0^{t+1}$, conditioned on the run-length and $\beta_t^{(r)} = Pr(r^t|x_0^{0:t})$ is the probability for each possible run-length given the observed data. Note that we know that, at any trial, there is a single true value for this variable $r^t$ and that $\beta_t^{(r)}$ thus represents the agent's inferred probability distribution over the run-length $r$. As a consequence, $\beta_t^{(r)}$ is scaled such that $\sum_{r \geq 0} \beta_t^{(r)} = 1$.

With these premises, we define the BBCP as a prediction / update cycle which connects nodes from the previous trial to that at the current trial. Indeed, we will *predict* the probability $\beta_t^{(r)}$ at each node, knowing either an initial prior, or its value on a previous trial. In particular, at the occurrence of the first trial, we know for certain that there is a switch and initial beliefs are thus set to the values $\beta_0^{(0)} = Pr(r^t = 0) = 1$ and $\forall r > 0, \beta_0^{(r)} = Pr(r^0 = r) = 0$. Then, at any trial $t > 0$, as we observe a new datum $x_0^t$, we use a knowledge of $\beta_{t-1}^{(r)}$ at trial $t - 1$, the likelihood $\pi_t^{(r)} = Pr(x_0^t|r^{t-1}, x_0^{0:t-1})$ and the transition probabilities defined by the generative model to predict the beliefs over all nodes:

$$\beta_t^{(r)} \propto \sum_{r^{t-1} \geq 0} \beta_{t-1}^{(r)} \cdot Pr(r^t|r^{t-1}) \cdot \pi_t^{(r)} \qquad (5)$$

In the computational graph, Eq 5 corresponds to a message passing from the nodes at time $t - 1$ to that at time $t$. We will now detail how to compute the transition probabilities and the likelihood.

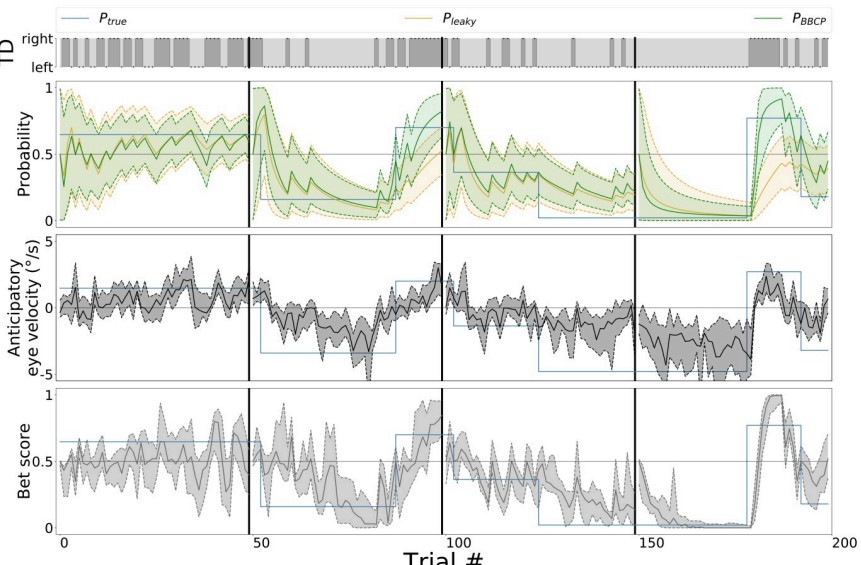

**Fig 2. Binary Bayesian Change-Point (BBCP) detection model.** *(A)* This plot shows a synthesized sequence of 13 events, either a leftward or rightward movement of the target (TD). Run-length estimates are expressed as hypotheses about the length of an epoch over which the probability bias was constant, that is, the number of trials since the last switch. Here, the true probability bias switched from a value of .5 to .9 at trial 7, as can be seen by the trajectory of the true run-length (blue line). The BBCP model tries to capture the occurrences of a switch by inferring the probability of different possible run-lengths. At any new datum (trial), this defines a Hidden Markov Model as a graph (trellis), where edges indicate that a message is being passed to update each node's probability (as represented by arrows from trial 13 to 14). Black arrows denote a progression of the run-length at the next step (no switch), while gray lines stand for the possibility that a switch happened: In this case the run-length would fall back to zero. The probability for each node is represented by the grey scale (darker grey colors denote higher probability) and the distribution is shown in the inset for two representative trials: 5 and 11. Overall, this graph shows how the model integrates information to accurately identify a switch and produce a prediction for the next trial (e.g. for $t = 14$). *(B)* On a longer sequence of 200 trials, representative of a trial block of our experimental sequence (see Fig 1A), we show the actual events which are observed by the agent (TD), along with the (hidden) dynamics of the true probability bias $P_{\text{true}}$ (blue line), the value inferred by a leaky integrator ($P_{\text{leaky}}$, orange line) and the results of the BBCP model in estimating the probability bias $P_{\text{BBCP}}$ (green line), along with .05 and .95 quantiles (shaded area). This shows that for the BBCP model, the accuracy of the predicted value of the probability bias is higher than for the leaky integrator. Below, we show the belief (as grayscales) for the different possible run-lengths. The green and orange line correspond to the mean run-length which is inferred, respectively, by the BBCP and leaky models: Note that in the BBCP, while it takes some trials to detect switches, they are in general correctly identified (transitions between diagonal lines) and that integration is thus faster than for the leaky integrator, as illustrated by the inferred value of the probability bias.

First, knowing that the data is generated by the Binary Switching model (see Eq 1), the run-length is either null at the moment of a switch, or its length (in number of trials) is incremented by 1 if no switch occurred:

$$
\begin{cases}
\text{if} & x_2^t = 1, \quad r^t = 0 \\
\text{else} & x_2^t = 0, \quad r^t = r^{t-1} + 1
\end{cases}
\tag{6}
$$

This may be illustrated by a graph in which information will be represented at the different nodes for each trial $t$. This defines the transition matrix $Pr(r^t | r^{t-1})$ as a partition in two exclusive possibilities: Either there was a switch or not. It allows us to compute the *growth probability* for each run-length. On the one hand, the belief of an increment of the run-length at the next trial is:

$$
\beta_t^{(r+1)} = \frac{1}{B} \cdot \beta_{t-1}^{(r)} \cdot \pi_t^{(r)} \cdot (1 - h)
\tag{7}
$$

where $h$ is the scalar defining the hazard rate. On the other hand, it also allows to express the change-point probability as:

$$\beta_t^{(0)} = \frac{1}{B} \cdot \sum_{r \geq 0} \beta_{t-1}^{(r)} \cdot \pi_t^{(r)} \cdot h \tag{8}$$

with $B$ such that $\sum_{r \geq 0} \beta_t^{(r)} = 1$. Note that $\beta_t^{(0)} = h$ and thus $B = \sum_{r \geq 0} \beta_{t-1}^{(r)} \cdot \pi_t^{(r)}$. Knowing this probability strength and the previous value of the prediction, we can therefore make a prediction for our belief of the probability bias at the next trial $t + 1$, prior to the observation of a new datum $x_0^{t+1}$ and resume the prediction / update cycle (see Eqs 4, 7 and 8).

Integrated in our cycle, we *update* beliefs on all nodes by computing the likelihood $\pi_t^{(r)}$ of the current datum $x_0^t$ knowing the current belief at each node, that is, based on observations from trials 0 to $t - 1$. A major algorithmic difference with the BCP model [58], is that here, the observed data is a Bernoulli trial and not a Gaussian random variable. The random variable $x_1^t$ is the probability bias used to generate the sequence of events $x_0^t$. We will infer it for all different hypotheses on $r^t$, that is, knowing there was a sequence of $r^t$ Bernoulli trials with a fixed probability bias in that epoch. Such an hypothesis will allow us to compute the distribution $Pr(x_0^{t+1} | r^t, x_0^{0:t})$ by a simple parameterization. Mathematically, a belief on the random variable $x_1^t$ is represented by the conjugate probability distribution of the binomial distribution, that is, by the beta-distribution $B(x_1^t; \mu_t^{(r)}, v_t^{(r)})$. It is parameterized here by its sufficient statistics, the mean $\mu_t^{(r)}$ and sample size $v_t^{(r)}$ (see Appendix for our choice of parameterization). First, at the occurrence of a switch (for the node $r^t = 0$) beliefs are set to prior values (before observing any datum): $\mu_t^{(0)} = \mu_{prior}$ and $v_t^{(0)} = v_{prior}$. By recurrence, one can show that at any trial $t > 0$, the sufficient statistics $(\mu_t^{(r)}, v_t^{(r)})$ can be updated from the previous trial following:

$$v_t^{(r+1)} = v_{t-1}^{(r)} + 1 \tag{9}$$

As a consequence, $\forall r, t; v_t^{(r)}$ is the sample size corrected by the initial condition, that is, $v_t^{(r)} = r + v_{prior}$. For the mean, the series defined by $\mu_t^{(r+1)}$ is the average at trial $t$ over the $r + 1$ last samples, which can also be written in a recursive fashion:

$$\mu_t^{(r+1)} = \frac{1}{v_t^{(r+1)}} \cdot \left( v_{t-1}^{(r)} \cdot \mu_{t-1}^{(r)} + x_0^t \right) \tag{10}$$

This updates for each node the sufficient statistics of the probability density function at the current trial.

We can now detail the computation of the likelihood of the current datum $x_0^t$ with respect to the current beliefs: $\pi_t^{(r)} = Pr(x_0^t | \mu_{t-1}^{(r)}, v_{t-1}^{(r)})$. This scalar is returned by the binary function $\mathcal{L}(r|o)$ which evaluates at each node $r$ the likelihood of the parameters of each node whenever we observe a counterfactual alternative outcome $o = 1$ or $o = 0$ (respectively right or left) knowing a mean bias $p = \mu_{t-1}^{(r)}$ and a sample size $r = v_{t-1}^{(r)}$. For each outcome, the likelihood of observing an occurrence of $o$, is the probability of a binomial random variable knowing an updated probability bias of $\frac{p \cdot r + o}{r + 1}$, a number $p \cdot r + o$ of trials going to the right and a number $(1 - p) \cdot r + 1 - o$ of trials to the left. After some algebra, this defines the likelihood as:

$$\mathcal{L}(r|o) = \frac{1}{Z} \cdot (p \cdot r + o)^{p \cdot r + o} \cdot ((1 - p) \cdot r + 1 - o)^{(1-p) \cdot r + 1 - o} \tag{11}$$

with $Z$ such that $\mathcal{L}(r|o = 1) + \mathcal{L}(r|o = 0) = 1$. The full derivation of this function is detailed

in Appendix. This provides us with the likelihood function and finally the scalar value $\pi_t^{(r)} = \mathcal{L}(r|x_0^t)$.

Finally, the agent infers at each trial the belief and parameters at each node and uses the marginal predictive probability (see Eq 4) as a readout. This probability bias is best predicted by its expected value $\hat{x}_1^{t+1} = Pr(x_0^{t+1}|x_0^{0:t})$ as it is marginalized over all run-lengths:

$$\hat{x}_1^{t+1} = \sum_{r \geq 0} \mu_t^{(r)} \cdot \beta_t^{(r)} \tag{12}$$

Interestingly, it can be proven that if, instead of updating beliefs with Eqs 7 and 8, we set nodes' beliefs to the constant vector $\beta_t^{(r)} = h \cdot (1-h)^r$, then the marginal probability is equal to that obtained with the leaky integrator (see Eq 2). This highlights again that, contrary to the leaky integrator, the BBCP model uses a dynamical model for the estimation of the volatility. Still, as for the latter, there is only one parameter $h = \frac{1}{\tau}$ which informs the BBCP model that the probability bias switches *on average* every $\tau$ trials. Moreover, note that the resulting operations (see Eqs 4, 7, 8, 11 and 12) which constitute the BBCP algorithm can be implemented *online*, that is, only the state at trial $t$ and the new datum $x_0^t$ are sufficient to predict all probabilities for the next trial. In summary, this prediction/update cycle exactly inverts the binary switching generative model and constitutes the Binary Bayesian Change-Point (BBCP) detection model.

**Quantitative analysis of the BBCP detection model.** We have implemented the BBCP algorithm using a set of Python scripts. This implementation provides also some control scripts to test the behavior of the algorithm with synthetic data. This strategy allows to qualitatively and quantitatively assess this ideal observer model against a ground truth before applying it on the trial sequence that was used for the experiments and ultimately comparing it to the human behavior. Fig 2A shows a graph-based representation of the BBCP estimate of the run-length for one instance of a short sequence (14 trials) of simulated data $x_0^t$ of leftward and rightward trials, with a switch in the probability bias of moving rightward occurring at trial 7 (see figure caption for a detailed explanation). Fig 2B, illustrates the predicted probability $\hat{x}_1^t$, as well as the corresponding uncertainty (the shaded areas correspond to .05 and .95 quantiles) when we applied respectively the BBCP (green curve) and the forgetful agent (orange curve) model to a longer sequence of 200 trials, characteristic of our behavioral experiments. In the bottom panel, we show the dynamical evolution of the belief on the latent variable (run-length), corresponding to the same sequence of 200 trials. The BBCP model achieves a correct detection of the switches after a short delay of a few trials.

Two main observations are noteworthy. First, after each detected switch, beliefs align along a linear ridge, as our model best estimate of the current run-length is steadily incremented by 1 at each trial until a new switch, and the probability $\hat{x}_1^t$ is predicted by integrating sensory evidence in this epoch: the model "stays". Then, we observe that shortly after a switch (an event that is hidden to the agent), the belief assigned to a smaller run-length smoothly increases while while the belief on the previous epoch decreases. At the trial for which the relative probability of the previous epoch is lower that that of the new, there is a transition to a new state: the model "goes". Such dynamic is similar to the slow / fast heuristic model proposed in other studies [62]. Second, we can use this information to readout the most likely probability bias and use it as a regressor for the behavioral data. Note that the leaky integrator model is implemented by the agent assuming a fixed-length profile (see orange line in Fig 2B), allowing for a simple comparison of the BBCP model with the leaky integrator. Again, we see that a fixed-length model gives qualitatively a similar output but with two disadvantages compared to the BBCP model, namely that there is a stronger inertia in the dynamics of the model estimates

and that there is no improvement in the precision of the estimates after a switch. In contrast, after a correct switch detection in the BBCP model, the value of the inferred probability converges rapidly to the true probability as the number of observations steadily increases after a switch.

In order to quantitatively evaluate the algorithm and following a similar strategy as [63], we computed an overall cost $\mathcal{C}$ as the negative log-likelihood (in bits) of the predicted probability bias, knowing the true probability and averaged over all $T$ trials:

$$\begin{cases} \mathcal{C} = \dfrac{1}{T}\sum_t \mathcal{C}(x_1^t, \hat{x}_1^t) \ \text{ with } \ \mathcal{C}(x_1^t, \hat{x}_1^t) = H(x_1^t, \hat{x}_1^t) - H(x_1^t, x_1^t) \\ \text{where } \ H(x_1^t, \hat{x}_1^t) = -x_1{}^t \log_2(\hat{x}_1^t) - (1 - x_1{}^t)\log_2(1 - \hat{x}_1^t) \end{cases} \quad (13)$$

The measure $\mathcal{C}(x_1^t, \hat{x}_1^t)$ explicitly corresponds to the average score of our model, as the Kullback-Leibler distance of $\hat{x}_1^t$ compared to the hidden true probability bias $x_1^t$. We have tested 100 trial blocks of 2000 trials for each read-out. In general, we found that the inference is better for the BBCP algorithm ($\mathcal{C} = 0.171 \pm 0.030$) than for the leaky integrator ($\mathcal{C} = 0.522 \pm 0.128$), confirming that it provides overall a better description of the data. Note that the only free parameter of this model is the hazard rate $h$ assumed by the agent (as in the fixed-length agent). Although more generic solutions exist [64–66], we decided as a first step to keep this parameter fixed for our agent, and evaluate how well it matches to the experimental outcomes at the different scales of the protocol: averaged over all observers, for each individual observer or independently in all individual trial blocks. In a second step, by testing different values of $h$ assumed by the agent but for a fixed hazard rate $h = 1/40$ in the Binary Switching model, we found that the distance given by Eq 13 is minimal for the true hazard rate used to generate the data. In other words, this analysis shows that the agent's inference is best for a hazard rate equal to that implemented in the generative model and which is actually hidden to the BBCP agent. This property will be important in a following section to validate the estimated hazard rate implicitly assumed by an individual participant on the basis of the set of responses given to the sequence of stimuli. As a summary, for each trial of any given sequence, we obtain an estimate of the probability bias assumed by the ideal observer and which we may use as a regressor. We will now present the analysis of this model's match to our experimental measures of anticipatory eye movements and explicit guesses about target motion direction.

## Anticipatory pursuit and explicit ratings

We used the binary switching model model to generate the (pseudo-)random sequence of the target's directions (the alternation of leftward/rightward trials) as the sequence of observations that were used in both sessions (see Fig 3). In the top panel of Fig 3, we show the actual sequence of binary choices (TD, leftward or rightward) of the Bernoulli trials. In the panel below, we compare the true value of the hidden probability bias $x_1$ (step-like blue curve), and the median predicted values using the leaky integrator ($P_{\text{leaky}}$, orange color) and BBCP model ($P_{\text{BBCP}}$, green color), along with the .05 to .95 quantile range (green shaded area), just as in Fig 2B. In the middle panel of Fig 3, we show the median (with the 0.25 ans 0.75 quantiles) anticipatory pursuit velocity (for details see Materials and methods) for the 12 participants, throughout a trial block of 200 trials of the experimental sequence. First, one can observe a trend in the polarity of anticipatory pursuit velocity to be negative for probability bias values below .5 and positive for values above .5. Comparing the raw anticipatory pursuit results with the BBCP agent predictions, it appears qualitatively that both traces evolve in good agreement. In particular, both curves unveil similar delays in detecting and taking into account a switch of the probability bias (while being hidden to the observers), reflecting the time (in the order of a few

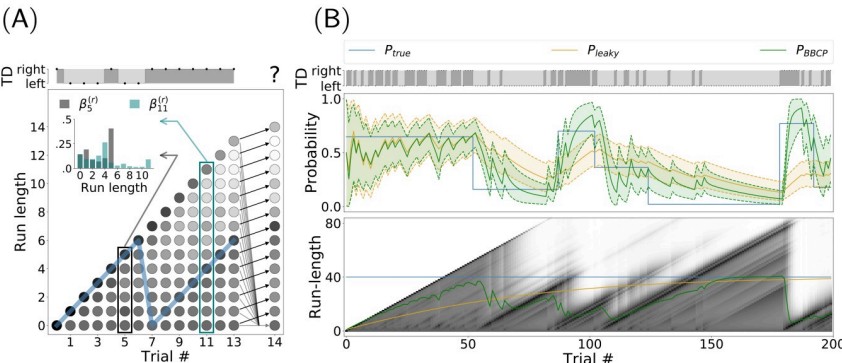

**Fig 3. Behavioral results, qualitative overview.** For one trial block of 200 trials, we compare the different model-estimated probabilities with respect to the behavioral results. The top row represents the sequence of target directions (TD) that were presented to observers and agents, as generated by the binary switching model (see Fig 1A). We show the evolution of the value of the (true) probability bias $P_{\text{true}}$ (blue line) which is hidden to observers and that is used to generate the TD sequence above. We have overlaid the results of the probability bias predicted with a leaky integrator ($P_{\text{leaky}}$, orange line) and with the BBCP model ($P_{\text{BBCP}}$, see Fig 2B, green line). Bottom two rows display the raw behavioral results for the $n$ = 12 observers, by showing their median (lines) and the .25 and .75 quantiles (shaded areas): First, we show the anticipatory pursuit eye velocity, as estimated right before the onset of the visually-driven pursuit. Below, we show the explicit ratings about the expected target direction (or *bet scores*). These plots show a good qualitative match between the experimental evidence and the BBCP model, in particular after the switches. Note that short pauses occurred every 50 trials (as denoted by vertical black lines, see main text), and we added the assumption in the model that there was a switch at each pause. This is reflected by the reset of the green curve close to the 0.5 level and the increase of the uncertainty after each pause.

trials) taken to integrate enough information to build up the estimation of a novel expectation about the probability bias value which parameterizes this Bernoulli trial. In general, results are more variable when the bias is weak ($p \approx .5$) than when it is strong (close to zero or one), consistent with the well-known dependence of the variance of a Bernoulli trial upon the probability bias ($\text{Var}(p) = p \cdot (1 - p)$). In addition, the precision (i.e. the inverse of the variance) of the inferred probability bias $\hat{x}_1$ increases in longer epochs, as information is integrated over more trials. As a result, the inferred probability as a function of time seems qualitatively to constitute a reliable regressor for predicting the amplitude of anticipatory pursuit velocity.

In addition, the explicit ratings for the next trial's expected motion direction (or *bet scores*, red curve in Fig 3) provided in the other experimental session seem to qualitatively follow the same trend. As with anticipatory pursuit, the series of the participants' bias guesses exhibits a positive correlation with the true probability bias: The next outcome of $x_0^t$ will in general be correctly inferred, as compared to a random choice, as reported previously [67]. Indeed, similarly to the amplitude of anticipatory pursuit velocity, we qualitatively compare in Fig 3 the trace of the bet scores with the probability bias $\hat{x}_1$ inferred by the BBCP model. Moreover, we observe again that a stronger probability bias leads to a lower variability in the bet scores, compared to bias values close to 0.5. Again, a (hidden) switch in the value of the bias is most of the time correctly identified after only a few trials. Finally, note that after every pause (black vertical bar in Fig 3), participants tended to favor unbiased guesses, closer to 0.5. We can speculate that this phenomenon could correspond to a spontaneous resetting mechanism of the internal belief on the probability bias and indeed, we can introduce such an assumption in the model as a reset of the internal belief after each pause. To conclude, the experiment performed in this session shows that the probability bias values that are explicitly estimated by participants are qualitatively similar to the implicit ones which supposedly underlie the generation of graded anticipatory pursuit.

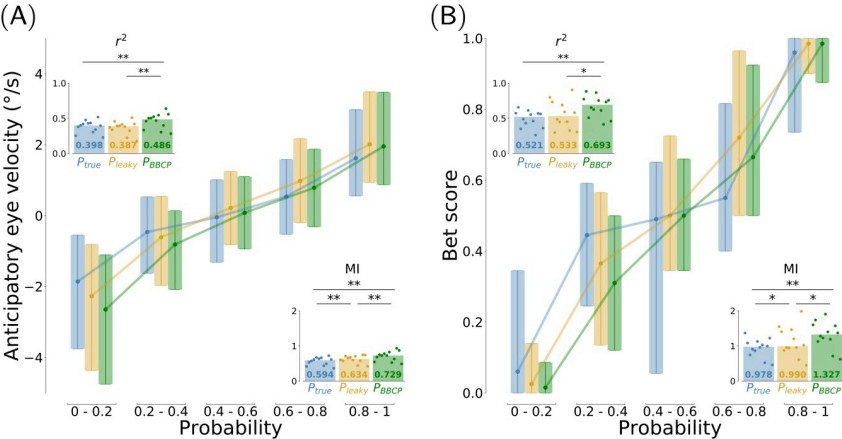

**Fig 4. Behavioral results, quantitative analysis across participants ($n = 12$).** To analyze the relation between these behavioral data with the predictions made by models, we first looked at the variability of all these measures conditioned on the predicted probability and gathered over 5 equal partitions of the [0, 1] probability segment. For the 12 participants, we collected an estimate of *(A)* the amplitude of anticipatory pursuit (aSPEM) and *(B)* the bet score value. As a regressor, we have used the true probability ($P_{\text{true}} = x_1^t$, blue color), and the probability bias estimates obtained with a leaky integrator ($P_{\text{leaky}}$, orange color) and by the BBCP model ($P_{\text{BBCP}}$, green color). We display these functional relations using an error-bar plot showing the median with .25 and .75 quantiles over the 5 partitions. This shows a monotonous dependency for both behavioral measures with respect to the probability, close to a linear regression, but with different strengths. Second, we summarize in insets quantitative measures of the strength of this dependence for each participant individually, by computing the squared Pearson correlation coefficient $r^2$ and the mutual information (MI). Dots correspond to these measures for each individual observer, while the bar gives the median value over the population. This confirms quantitatively that for both experimental measures, there is a strong statistical dependency between the behavioral results and the prediction of the BBCP model, but also that this dependency is significantly stronger than that obtained with the true probability and with the estimates obtained with the leaky integrator (stars denote significative differences, see text for details).

Quantitatively, we now compare the experimental results with the value of the probability bias $\hat{x}_1$ predicted by the leaky and BBCP algorithms. Compiling results from all participants, we have plotted in Fig 4 the anticipatory pursuit velocity (panel A) and the bet scores (panel B) as a function of the predicted probability biases. In a first analysis, all trials from all participants were pooled together and we show this joint data as an error bar plot as computed for 5 equal partitions of the [0, 1] probability segment showing the median along with the .25 and .75 quantiles. As a comparison, the same method was applied to the true value $P_{\text{true}}$ and to the estimate obtained by the leaky integrator $P_{\text{leaky}}$. We remind here that the true value of the probability bias was coded at the second layer of the binary switching generative model and is hidden both to the agents and to the human observers. Qualitatively, as we can see in Fig 4A, the predicted probability bias is linearly correlated with the anticipatory pursuit velocity and this dependence is stronger with the the probability bias predicted by the leaky and BBCP algorithms (respectively $P_{\text{leaky}}$ and $P_{\text{BBCP}}$). In a second analysis, we quantitatively estimated the squared Pearson correlation coefficient and the mutual information between the raw data and the different models, both as computed on the whole data or for each observer individually (see insets in Fig 4). The respective values for the whole dataset ($r^2 = 0.486$ and $MI = 0.729$) and across participants ($r^2 = 0.459 \pm 0.104$ and $MI = 0.707 \pm 0.134$) are slightly higher than that found by [48] and [11] for anticipatory pursuit measures gathered across experimental trial blocks with fixed direction biases and significantly better than that estimated with the true probability ($r^2 = 0.381 \pm 0.083$ with $p = 0.002$ and $MI = 0.562 \pm 0.107$ with $p = 0.002$) and for that estimated by the leaky-integrator model ($r^2 = 0.366 \pm 0.089$ with $p = 0.002$ and

$MI = 0.622 \pm 0.102$ with $p = 0.004$) see inset). Note that $p$-values were obtained from the Wilcoxon signed-rank test.

A similar analysis illustrates the relationship between the model-estimated probability bias and the rating value, or bet score, about the expected outcome, which was provided at each trial by participants and is shown in Fig 3. Similarly to the anticipatory pursuit velocity, the rating values are nicely correlated with the probability bias given by the model, as quantified by the squared Pearson correlation coefficient and mutual information across participants ($r^2 = 0.670 \pm 0.145$ and $MI = 1.312 \pm 0.364$). Importantly, this value is again higher for the BBCP model than for the leaky integrator ($r^2 = 0.551 \pm 0.19$ with $p = 0.018$ and $MI = 1.117 \pm 0.409$ with $p = 0.028$), or with the true probability ($r^2 = 0.490 \pm 0.114$ with $p = 0.002$ and $MI = 0.940 \pm 0.255$ with $p = 0.002$). Further notice that, in order to account for some specific changes observed in the behavioral data after the short pauses occurring every 50 trials, we added the assumption that there was a switch at each pause. However, removing this assumption did not significantly change the conclusions about the match of the model compared to $P_{\text{true}}$ or $P_{\text{leaky}}$ both for eye movements ($P_{\text{BBCP}}$: $r^2 = 0.452 \pm 0.101$ and $MI = 0.712 \pm 0.125$, $P_{\text{leaky}}$: $r^2 = 0.305 \pm 0.077$ with $p = 0.002$ and $MI = 0.577 \pm 0.096$ with $p = 0.003$; $P_{\text{true}}$: $r^2 = 0.381 \pm 0.083$ with $p = 0.002$ and $MI = 0.562 \pm 0.107$ with $p = 0.002$) and the bet experiment ($P_{\text{BBCP}}$: $r^2 = 0.652 \pm 0.142$ and $MI = 1.255 \pm 0.349$, $P_{\text{leaky}}$: $r^2 = 0.425 \pm 0.158$ with $p = 0.002$ and $MI = 0.966 \pm 0.300$ with $p = 0.002$; $P_{\text{true}}$: $r^2 = 0.490 \pm 0.114$ with $p = 0.002$ and $MI = 0.940 \pm 0.255$ with $p = 0.002$). To conclude, we deduce that the dynamic estimate of the probability bias produced by the BBCP model is a powerful regressor to explain both the amplitude of anticipatory pursuit velocity and the explicit ratings of human observers experiencing a volatile context for visual motion.

## Analyzing inter-individual differences

So far, we have presented the qualitative behavior of individual participants and have reported the quantitative analysis of the data for the fit between experimental and model-inferred estimates of the hidden probability bias. For instance, the experimental measures for the population of 12 participants in Fig 3, support the qualitative match between behavioral data and model predictions, which we then confirmed quantitatively on the whole group of participants. It is important to note that no model fitting procedure was used so far, but only the direct match of the prediction from the BBCP-model resulting from the sequence of binary target directions which were also presented to the human participants, as shown in Fig 2B. Nevertheless, we observed that in both sessions the qualitative match between model and data varied across participants. This was best characterized by differences in the variability of the responses, but also, for instance, by the different characteristic delays after a switch. This reflects the spectrum of individual behavioral choices between exploitation versus exploration [60]. As a consequence, we were interested in characterizing these individual preferences for each individual participant, and potentially to investigate whether this preference co-varied across the two experimental sessions (i.e. across implicit vs explicit response modalities). Crucially, we have seen that the BBCP model is controlled by a single parameter, the hazard rate, or equivalently by its inverse, the characteristic time $\tau$. Also, we have shown that knowing an observed sequence of behavioral responses, we could fit the value of $h$ which would best explain the observations, as quantified by the squared Pearson correlation coefficient or by the mutual information. Thus, by extracting the best-fit parameters for each participant and experimental session, we expect to better understand the variety of inter-individual differences.

Hence, we have fitted the sequence of behavioral responses generated by each participant and for each experimental session, with the predicted probability bias predicted by the BBCP

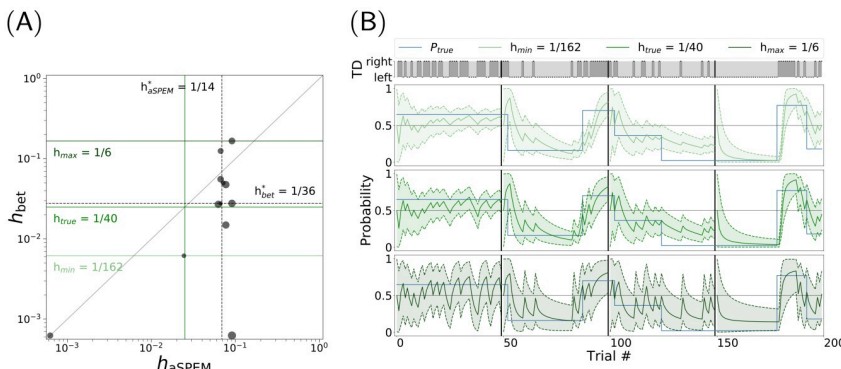

**Fig 5. Analysis of inter-individual differences.** *(A)* We analyzed the behavior of the $n = 12$ participants individually, by searching for each participant the best value of the model's single free parameter, the hazard rate $h$. Estimates were performed independently on both experiments, such that we extracted different estimates of $h_{aSPEM}$ and $h_{bet}$ respectively for the anticipatory pursuit velocity and the rating value. The dots correspond to independent estimates of the hazard rate for each individual participant are shown as dots, while the radius is proportional to the squared Person's correlation coefficient. This plot shows that best fit hazard rates have a median value of $h^*_{aSPEM} = \frac{1}{14}$ and $h^*_{bet} = \frac{1}{36}$. The values are in general higher than the ground truth (blue line), and in general higher for eye movements (below the diagonal). Note that the dispersion of hazard-rate best-fit estimates is narrower for the eye movement session than for the bet experiment. Such an analysis may suggest that participants ultimately have different mechanisms at the implicit (anticipatory pursuit) and explicit (ratings) levels for guiding their tendency of exploitation versus exploration. *(B)* To illustrate the models corresponding to these best-fitted values of the hazard rates, we show the predicted probability to the same sequence of TDs, with the lowest $(\frac{1}{162})$, optimal $(\frac{1}{40})$ and highest $(\frac{1}{6})$ hazard rates (respectively from top to bottom).

model with different values of its only free parameter, the hazard rate. To avoid any possible bias from the fitting procedure, we tested 1600 linearly spaced values of $\tau$ from 1 to 1600 trials. For each, we computed the correlation coefficient with the responses of the BBCP model parameterized by the value of the hazard rate $h = \frac{1}{\tau}$. We then extracted different estimates of $h_{aSPEM}$ and $h_{bet}$, respectively for anticipatory pursuit and the rating scale, by choosing the hazard rate value corresponding to that with maximal correlation coefficient. The scatter plot of the best fit values for each individual is shown in Fig 5. This figure suggests, in the first place, that there is some variability in the best fitted value of the hazard rate in both sessions. Overall, the value of correlation coefficient of the best fit hazard rate was slightly higher than that computed in Fig 3 with $r^2 = 0.471 \pm 0.109$ for the eye movement session and $r^2 = 0.691 \pm 0.152$ for the rating scale session. A part of the variability in the estimated hazard rates comes from the limited length of the data blocks, while another part is due to intra-individual and inter-individual variabilities. Overall, the median (with 25% and 75% quantiles) are $h_{aSPEM} = 0.069$ (0.065, 0.080) for the anticipatory pursuit session and $h_{bet} = 0.027$ (0.012, 0.051) for the rating scale. We observe that these values are close to the (hidden) ground truth value ($h = 1/40 = 0.025$) used to generate the sequence. In addition, the best-fit hazard rate value is higher for anticipatory pursuit compared to the true value and the rating scale measures. As an interim summary, this analysis reveals that relaxing the free parameter of the BBCP model improves the match of the model to the behavioral data, and that individual best-fitting hazard-rates are variable, especially for the Bet task. Future work might provide important insight about the analysis of these inter-individual differences in terms of each participant's preference for exploration versus exploitation across different cognitive tasks.

The distribution of best-fitted values for each individual participant seemed to qualitatively cluster, but the dataset is still insufficiently large to support the significance of such observation at a quantitative level. Moreover, there is a difference in the distribution of observed hazard

rates in both experiments. Indeed, we observed that the marginal distribution for each session is different, with the distribution in the anticipatory pursuit session being narrower than that observed for the rating scale session. In particular, we also observed the same behavior for each trial block independently, suggesting that the origin of this variability mainly comes from inter-individual variability. Second, there is an apparent lack of correlation between the explicit and the implicit estimates of the hazard rate, yet we would need more empirical evidence to prove that this originates from the experimental setup or rather by separate processing of volatility. Such an analysis would suggest that even though the predictive processes at work in both sessions may reflect a common origin for the evaluation of volatility, this estimation is then more strongly modulated by individual preferences when a more explicit cognitive process is at stake.

## Discussion

The capacity to adapt our behavior to the environmental regularities has been investigated in different research fields, from motor priming and sensory adaptation to reinforcement learning, machine learning and economics. Several studies have aimed at characterizing the typical time scale over which such adaptation occurs. However, the pattern of environmental regularities could very well change in time, thereby making a fixed time-scale for adaptation a suboptimal cognitive strategy. In addition, different behaviors are submitted to different constraints and respond to different challenges, thus it is reasonable to expect some differences in the way (and time scales) they adapt to the changing environment. This study is an attempt to address these crucial open questions. We have taken an original approach, by assuming a theoretically-defined volatility in the properties of the environment (in the specific context of visual motion tracking) and we have developed an optimal inferential agent, which best captures the hidden properties of the generative model solely based on the trial sequence of target motion. We have then compared the optimal agent's prediction, as well as a more classical *forgetful* agent, to two sets of behavioral data, one rooted in the early oculomotor network underlying anticipatory tracking, and the other related to the explicit estimate of the likelihood of a future event. Our results point to a flexible adaptation strategy in humans, taking into account the volatility of the environmental statistics. The time-scale of this dynamic adaptive process would thus vary across time, but it would also be modulated by the specific behavioral task and by inter-individual differences. In this section we discuss the present work and its implications in view of the existing literature and some general open questions.

### Measuring adaptation to volatile environments

The time-varying statistical regularities that characterize the environment are likely to influence several cognitive functions. In this study, we have made the choice to focus on a simple and probably mostly unconscious motor behavior (anticipatory pursuit), as well as on the explicit rating of expectation for the forthcoming motion direction. In contrast, we have not addressed the question of whether and how statistical learning affects visual motion perception throughout our model-generated volatile sequences. In an empirical context similar to ours, Maus et al [10] have recently shown that perceptual adaptation for speed estimation occurs concurrently to priming-based anticipatory pursuit throughout a sequence of motion tracking trials with randomly varying speed. They actually found a robust *repulsive* adaptation effect, with perceptual judgements biased in favor of faster percepts after seeing stimuli that were slower and *vice-versa*. Concurrently, these authors also found a positive effect on anticipatory pursuit, with faster anticipation after faster stimuli, somehow in agreement with the adaptive properties of anticipatory pursuit that we report here. Moreover, they quantified the trial-

history effects on anticipatory pursuit and speed perception by fitting a fixed-size memory model similar to our forgetful agent. They found that anticipatory pursuit and speed perception change over different time scales, with the priming effects being maximized for short-term stimulus history (around 2 trials) and adaptation for longer stimulus history, around 15 trials. Their main conclusion was that perceptual adaptation and oculomotor priming are the result of two distinct readout processes using the same internal representation of motion regularities. Note that both these history lengths can be considered short in comparison to the several hundreds of trials that are commonly used in psychophysics and sensorimotor adaptation studies and that, similar to the present study, the inferred characteristic times are even shorter for the buildup of anticipatory eye movements. However, it is also important to note that in the study by Maus et al [10], the generative model underlying the random sequence of motion trials was different and much simpler than in the present study: In particular the role of environmental volatility was not directly addressed there. This makes a direct comparison between their results and ours difficult beyond a qualitative level.

In spite of a multitude of existing studies investigating the dynamics of sequential effects on visual perception (see for example [5, 7]), only few of them have directly addressed the role of the environmental volatility on the different behavioral outcomes. Meyniel et al [24] have compared the predictions of different models, featuring a dynamic adaptation to the environment's volatility (equivalent to our *forgetful agent model*) versus a fixed belief model, on five sets of previously acquired data, including reaction time, explicit reports and neurophysiological measures. Interestingly, they concluded that the estimation of a time-varying transition probability matrix constitutes a core building block of sequence knowledge in the brain, which then applies to a variety of sensory modalities and experimental situations. Consequently, sequential effects in binary sequences would be better explained by learning and updating transition probabilities compared to the absolute item frequencies (as in the present work) or the frequencies of their alternations. The critical difference lies in the content of what is learned (transition probabilities versus item frequencies) in an attempt to capture human behavior. Rather than on transition probabilities, here we focused on the analysis and modeling of human behavior as a function of the frequency of presentation (and its fluctuations in time) of a given event in a binary sequence of alternating visual motion direction. We can speculate that different statistics can play different roles depending on the context, but altogether the study by Meyniel et al [24] and the present one converge to highlight the importance of a dynamic estimate of the hierarchical statistical properties of the environment for efficient behavior. There are also other limits to the agent that we have defined. In this study, we assume that data is provided as a sequence of discrete steps. A similar approach using a Poisson point process allows to extend our model to the continuous time domain, such as addressed by Radillo et al [68]: In their experiments, the authors analyzed the licking behavior of rats in a dynamic environment. The generalization to the time-continuous case is beyond the scope of our current protocol, but it would consist in a natural extension of it to more complex and ecological settings.

Our results demonstrate that the BCCP model is relatively good in mimicking the adaptive changes of both (implicit) anticipatory eye movements and (explicit) ratings of direction expectation in a volatile context. However, these two different behavioral measures, the implicit and the explicit one, are not correlated across individuals. This observation is certainly worth deeper investigation in the future as it raises doubts on the existence of a unique hierarchical system for probabilistic inference. The distinction between implicit and explicit processes in the adaptation to a volatile environment has also been addressed by previous work, especially in the field of statistical learning for language processing (see for example [69, 70]). More related to the present study, Wu et al [71] compared a classical economic decision task

with a motor decision task: they found that participants were more risk seeking in the motor task compared to the first one. In addition, Souto et al [72] have recently reported a lack of correlation between the rate of oculomotor adaptation to unexpected jumps of the visual target and the perceptual uncertainty estimated through an explicit jump direction-discrimination experiment. Finally, the degree of explicitness of the information provided to the participants may also play a role in the context of probabilistic learning. In a task similar to ours, where the behavioral choice was not specifically associated to a reward schedule, Santos and Kowler [51] found large similarities but also some differences in the anticipatory eye movements depending on how the information about the probability bias was conveyed, namely through the simple presentation of a biased sequence versus an explicit probability-cueing procedure. In the first condition, the authors reported a weak non-linearity in the dependence of anticipatory pursuit upon the probability of motion direction, yielding an overweight of the extreme values of probability. In contrast, an opposite non-linearity (underweight) was observed when the target direction was visually-cued with a given probability of validity. Note that in our data, we have not found consistent evidence suggesting a clear non-linearity in either sense. Further work is needed to disentangle the possible specificities (e.g. non linearities, also broadly reported in the economic literature, such as a generic aversion to risk [73]) and the general inter-trial and inter-individual correlations across different tasks and different experimental measures of the cognitive adaptation to the environmental volatility.

## Hierarchical Bayesian inference in the brain

When we perceive the physical world, make a decision or take an action to interact with it, our brain must deal with an ubiquitous property of it, uncertainty. Uncertainty can arise at different levels and be structured around different characteristic time scales. The theoretical framework of Bayesian probabilistic inference, which provides a formal account for the role of uncertainty at multiple levels, has become very popular as a benchmark of optimal behavior in perceptual, sensorimotor and cognitive tasks [74] and, more generally, as a unified framework for studying the brain [75]. Importantly, plausible hypotheses about the implementation of Bayesian computations —or approximations of them— in the activity of neuronal populations have been proposed [76–78]. However, one should be careful when evaluating the quality of fit of Bayesian inference models for behavioral data, and avoid any over-interpretation of the results. This kind of model fitting aims at evaluating the adequacy of a specific generative inferential model, not of the probabilistic calculus in its detailed implementation. Still, there is actually a common confusion around the idea of a "Bayesian brain", and we believe that the challenge here is not to validate the hypothesis that the brain implements or not the Bayes' theorem, or a more complex hierarchical combinations of inferential computations, but rather to test hypotheses about the different generative models that agents may use.

The way expectations act on cognitive processes in general has been investigated in a wide range of domains such as predictive coding [79], active inference [75], motor control [80] and reinforcement learning [11, 60, 65]. Non-stationary observations can also explain why both local and global effects emerge and why local effects persist in the long run even within purely random sequences [28, 81]. This constant update of a general belief on the world can be a consequence of the constant attempt to learn the non-stationary structure of the environment that can change at unpredictable times [81]. Many studies have actually already pointed out the brain's ability to apprehend non-stationary states in the environment [67, 82]. The relatively strong correlation between model predictions and data that we have found in this study is surprising at a first sight as the epochs with constant probability bias (between two switches) have

random lengths, and participants have to adapt to such a volatile environment. However, adaptivity to a volatile environment is one of the most exquisite human skills: When faced with some new observations, the observer has to constantly adapt his/her response to either exploit this information by considering that this observation belongs to the same context of the previous observations, or to explore a novel hypothesis about the context. This compromise is one of the crucial components that we wished to explore and which is well captured by the BBCP model. In particular, the model predicts different aspects of the experimental results, from the variability as a function of the inferred probability, to the dynamics of the behavior following a (hidden) switch. Future work will be needed to address the amplitude and dynamics of modulations of visual perception and other cognitive functions in a model-based volatile environment like the one we formally defined in this study, and to compare them to other implicit and explicit behavioral measures (like anticipatory eye movements and explicit expectation ratings).

The great interest of understanding why and how humans adapt to the fluctuations of the hierarchical probabilistic context is further highlighted by the fact that such adaptivity may deviate in some pathological disorders, such as schizophrenia [4, 83], or across the natural variability of autistic traits [84]. While it was not our original objective, we have analyzed in this study the individual best-fit parameters (hazard rates) of the BCCP model: despite a consistent variability of such parameters across trial blocks of the experiment, we highlighted some noteworthy tendencies for participants to cluster around specific properties of the dynamic adaptation to a volatile probabilistic environment. Most important, this analysis corroborates and strengthens some recent attempts to realize a *computational phenotyping* of human participants. However, more extensive studies should be conducted to be able to quantitatively titrate inter-individual tendencies and possibly their relation to traits of personality.

## Conclusions

- We have developed a Bayesian model of an agent estimating the probability bias of a volatile environment with changing points (switches), such that the agent may decide *to stay* on the current hypothesis about the environment, or *to go* for a novel one. This allows to dynamically infer the probability bias across time and directly compare model predictions and experimental data, such as measures of adaptive human behavior.

- We applied such a framework to the case of a probability bias in a visual motion task where we manipulated the target direction probability. We observed a good match between anticipatory smooth eye movements and the predictions of the model, replicating previous findings and providing a novel solid theoretical framework for them [11, 48, 51].

- We also found a good match between model predictions and the explicit rating of the expected target motion direction, a novel result suggesting that this model captures some of the brain computations underlying expectancy based motion prediction, at different cognitive levels.

- Finally, we found that the experimental data of each different participant matched to different types of belief about the volatile environment, some being more or less conservative than others. Interestingly, each of the two experiments (anticipatory eye movements and explicit rating) provided different distributions, opening the perspective for future *computational phenotyping* using such a volatile setting.

## Materials and methods

### Participants, visual stimuli and experimental design

Twelve observers (29 years old ±5.15, 7 female) with normal or corrected-to-normal vision took part in these experiments. They gave their informed consent and the experiments had received ethical approval from the Aix-Marseille Ethics Committee (approval 2014-12-3-05), in accordance with the declaration of Helsinki.

Visual stimuli were generated using PsychoPy 1.85.2 [85] on a Mac running OS 10.6.8 and displayed on a 22" Samsung SyncMaster 2233 monitor with $1680 \times 1050$ pixels resolution at 100 Hz refresh rate. Experimental routines were also written using PsychoPy and controlled the stimulus display (see Fig 1). Observers sat 57 cm from the screen in a dark room.

The moving target used in our experiments was a white ring (0.35˚ outer diameter and 0.27˚ inner diameter) with a luminance of 102 $cd/m2$ that moved horizontally on a grey background (luminance 42 $cd/m^2$). Each trial started with a central fixation point displayed for a random duration drawn from a uniform distribution ranging between 400 and 800 ms. Then a fixed-duration 300 ms gap occurred between the offset of the fixation point and the onset of the moving target. The target was then presented slightly offset from the fixation location (*step-ramp* paradigm [86]), either to the right or to the left, and immediately started moving horizontally toward the center at a constant speed of 15˚/*s*, for 1000 ms. The probability *p* of rightward motion trials was a time-varying random variable which was constant within an epoch of the sequence of a given random size (see main text for the description of the generative model).

The paradigm included two experimental sessions performed on two distinct (in general consecutive) days by each participant. The two sessions involved the presentation of the same sequence of trials, while collecting a different behavioral response: explicit rating judgments in the first session (the *bet* experiment), and eye movement recordings in the second session. Asked after the experiment, no observer noticed that the same pseudo-random sequence of target directions was used in both experiments.

### Eye movements experiment

Eye movements were recorded continuously with an eye tracking system (Eyelink 1000, SR Research Ltd., sampled at 1000 Hz), using the Python module Pylink 0.1.0 provided by PsychoPy. Horizontal and vertical eye position data were transferred, stored, and analyzed offline using programs written using Jupyter notebooks. The data analyses were implemented using the Python libraries numpy, pandas and pylab. All the scripts for data analysis, as well as for stimulus presentation, data collection, and preparation of figures are available at https://github.com/chloepasturel/AnticipatorySPEM. To minimize measurement errors, the participant's head movements were restrained using a chin and forehead rest, so that the eyes in primary gaze position were directed towards the center of the screen. In order to enforce accuracy in gaze position and tracking, we implemented an automatic procedure of fixation control. If the distance between the gaze position and the central fixation point during the fixation epoch exceeded 2˚ of visual angles, the fixation point started flickering and the counter for the fixation duration was reset to 0.

The recorded horizontal and vertical raw gaze position data were numerically differentiated to obtain velocity measures. We adopted an automatic conjoint acceleration and velocity threshold method (the default saccade detection implemented by SR Research) to detect ocular saccades. Saccades and eye-blinks were excluded from eye velocity traces (and replaced by *Not-a-Number* values in the numerical arrays) before trial averaging and data fitting for the extraction of the oculomotor parameters of interest. In order to extract the relevant parameters

of the oculomotor responses, we developed new tools based on a best-fitting procedure of pre-defined oculomotor patterns and in particular the typical smooth pursuit velocity profile that was recorded in our experiment. A piecewise-defined function was fitted to the different phases of the eye velocity traces: a constant function during fixation, a ramp-like linear function during smooth pursuit anticipation, an increasing sigmoid function during the initiation of visually-guided smooth pursuit, reaching its saturating value during the pursuit steady-state. This analysis was applied to each trial individually and it allowed in particular to estimate the velocity of anticipatory pursuit as the best-fit value of the modeled eye velocity at the moment where the visually-guided pursuit is initiated. Note that this method for estimating anticipatory velocity led to qualitatively identical results to the estimation of the mean eye velocity within an arbitrary temporal window of anticipation, a more classical method that we implemented for instance in a previous study [11]. Some trials were excluded from the analysis as the proportion of missing data-points, due to eye blinks or saccades was considered too large, namely when the missing data exceeded 45 ms during the gap or one third of the total target motion epoch (4.36% of all trials). In addition, trials were also excluded when the eye-movement fitting procedure did not converge, after visual inspection, to a satisfactory match with the data (3.25% of all trials). The python scripts used to analyze eye movements are available at https://github.com/invibe/ANEMO.

## The Bet experiment

The aim of the Bet experiment was to collect data related to the individual explicit estimates of the probability for the next outcome of a target motion direction. At the beginning of each trial, before the presentation of the moving target, participants had to answer to the question *"How sure are you that the target will go left or right"*. This was performed by adjusting a cursor on the screen using the mouse (see Fig 1C). The cursor could be placed at any point along a horizontal segment representing a linear rating scale with three ticks labeled as *"Left"*, *"Right"* (at the extreme left and right end of the segment respectively), and *"Unsure"* in the middle. Participants had to validate their choice by clicking on the mouse left-button and the actual target motion was shown thereafter. The rationale to collect rating responses on a continuous scale instead of a simple binary prediction (Right/Left) was to be able to infer the individual estimate of the direction bias at the single trial scale (in analogy to the continuous interval for the anticipatory pursuit velocity). We called this experiment the *"Bet"* experiment, as participants were explicitly encouraged to make reasonable rating estimates, just like if they had to bet money on the next trial outcome. Every 50 trials, a *"score"* was displayed on the screen, corresponding to the proportion of correct direction predictions (Right or Left of the *"Unsure"* tick) and weighted by the confidence attributed to each answer (the distance of the cursor from the center).

## Supporting information

**S1 Algorithm. Detailed explanation of the BBCP algorithm.**
(PDF)

## Acknowledgments

We thank Doctor Jean-Bernard Damasse, Guillaume S Masson and Professor Laurent Madelain for insightful discussions.

## Author Contributions

**Conceptualization:** Anna Montagnini, Laurent Udo Perrinet.

**Data curation:** Chloé Pasturel.

**Formal analysis:** Chloé Pasturel, Anna Montagnini, Laurent Udo Perrinet.

**Funding acquisition:** Anna Montagnini, Laurent Udo Perrinet.

**Investigation:** Anna Montagnini, Laurent Udo Perrinet.

**Methodology:** Chloé Pasturel, Anna Montagnini, Laurent Udo Perrinet.

**Project administration:** Chloé Pasturel, Laurent Udo Perrinet.

**Resources:** Anna Montagnini, Laurent Udo Perrinet.

**Software:** Laurent Udo Perrinet.

**Supervision:** Anna Montagnini, Laurent Udo Perrinet.

**Validation:** Anna Montagnini, Laurent Udo Perrinet.

**Visualization:** Chloé Pasturel, Anna Montagnini, Laurent Udo Perrinet.

**Writing – original draft:** Anna Montagnini, Laurent Udo Perrinet.

**Writing – review & editing:** Chloé Pasturel, Anna Montagnini, Laurent Udo Perrinet.

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
