## [Decision Letter · Decision Letter 0]

23 Dec 2019

Dear Dr Perrinet,

Thank you very much for submitting your manuscript, 'Humans adapt their anticipatory eye movements to the volatility of visual motion properties', to PLOS Computational Biology. As with all papers submitted to the journal, yours was fully evaluated by the PLOS Computational Biology editorial team, and in this case, by independent peer reviewers. The reviewers appreciated the attention to an important topic but identified some aspects of the manuscript that should be improved.

We would therefore like to ask you to modify the manuscript according to the review recommendations before we can consider your manuscript for acceptance. Your revisions should address the specific points made by each reviewer and we encourage you to respond to particular issues Please note while forming your response, if your article is accepted, you may have the opportunity to make the peer review history publicly available. The record will include editor decision letters (with reviews) and your responses to reviewer comments. If eligible, we will contact you to opt in or out.raised.

- Supporting Information uploaded as separate files, titled 'Dataset', 'Figure', 'Table', 'Text', 'Protocol', 'Audio', or 'Video'.

We hope to receive your revised manuscript within the next 30 days. If you anticipate any delay in its return, we ask that you let us know the expected resubmission date by email at ploscompbiol@plos.org.

Sincerely,

Wolfgang Einhäuser

Deputy Editor

PLOS Computational Biology

[LINK]

Reviewer's Responses to Questions

**Comments to the Authors:**

Reviewer #1: The authors use anticipatory pursuit eye movements and explicit judgements to study how humans adapt to the volatility of their environment (i.e. how the probability of specific events changes; foraging for food could give an ecologically relevant example). Whereas anticipatory pursuit and other behaviours have been studied rather extensively to investigate the integration of probability in decision-making there has been little attention being paid to higher-level statistics. The authors put forth a Bayesian model for explaining adaptation to the latter, extending considerably the complexity of current models. The experimental paradigm is a simple task in which the observer sees leftward and rightward motion of a dot with varying probabilities. Those probabilities are fixed within a block of trials and this probabilities change at random run lengths (unbeknown to the participant).

The article centres upon contrasting a new Bayesian model predicting probability transitions to a simple leaky integrator model. They then compare the appropriateness of the model in explaining behaviour (anticipatory pursuit, i.e. the supposedly implicit eye movement that anticipates target motion, and the explicit "bets" on whether the target moves leftward or rightward).

If find the paper an interesting addition to the rather vast literature on sequence learning with implicit and explicit measures. The authors show indeed that the Bayesian model outperforms the leaky integrator model, providing a useful and novel description of sequence learning.

Major points

I feel the writing could be much more concise and terms could be used more consistently. For instance, lines 773-809 of the discussion seem entirely gratuitous to me. Overall, the readership may be put off by repetition and at times an overly abstract argumentation. I also feel that the mathematical explanation of the model could be made clearer, perhaps by using different symbols rather than superscript to differentiate between different estimates.

While the discussion lacks concision, on the other hand the authors take little stock of their findings. For instance, the lack of correlation between explicit and implicit measures when it comes to estimates of parameter h (a very similar point was made here: https://www.frontiersin.org/articles/10.3389/fnhum.2016.00227/full), which they call hazard rate. Also, I was expecting some discussion of something that is rather conspicuous in Figure 4A. The eye movements seem to match the true probability better than both the leaky and Bayesian model. It would seem that the Bayesian model doesn't fare well to explain the eye movements. I am then left wondering what further information the user can use to beat the optimal observer?

It seems odd not to cite the work of Collins and Barnes on anticipatory pursuit with different probability blocks: https://www.jneurosci.org/content/29/42/13302.short

Regarding the mathematical interpretation I am not sure that hazard rate is a good term for the term h, which in a leaky-integrator model is more known as the leaking rate or forgetting rate. The hazard rate refers to something different. I am not sure whether I am missing something.

Minor and more specific comments

A matter of taste perhaps: Acronyms often save time for the writer but not the reader, also I find aSPEM are rather ugly-sounding acronym. "anticipatory pursuit", isn't much longer.

lines 61-63 the link to the measles outbreak could be made explicit.

Consider removing needless words: "As such,", "in all generality," ...

Paragraph 76-91 is rather obscure to me, it requires some more explanation. The link to adaptation and priming is probably misleading as the measure corresponds rather to a prediction of what happens next, those two effects are typically tested differently (a reduction of sensitivity and an increase in sensitivity)

line 79: "yielding a"

line 82: not clear how adaptation "favors spatial stability of the stimulus"

line 86: quite obscure

The statement line 145 needs to be substantiated. I am not aware of a study testing this that anticipatory pursuit is unconscious, although work on saccades (capture) would suggest so based on the time-scales involved. This points appears rather central in the motivation of the study (contrast implicit and explicit mechanisms), therefore it needs more consideration.

line 160: coherent "with"

line 192: Is it meant to be "volatility" and not "variability"?

line 195-197 the term trial block may lead to some confusion, as it is meant to be the sequence of trials with the same probability but also commonly the number of trials in a full sequence.

line 318: the reference should be included when you first mention this heuristic, otherwise it feels like you refer to a new one.

I can't make much sense of 324-326. The point that Equation 3 assumes a constant weight is also obscure to me. Doesn't a leaky-integrator amount to weighting past trials depending on trial number? This needs to be clarified.

Figure 2: I take the blue line in A to represent the black lines being mentioned.

Fig 4. An indication of n on the figure would be welcome (I assume it is n observers). "For all participants and for all trials" is an ambiguous description. Was it one estimate per participant, averaging all trials? or were all trials pooled together disregarding participants?

Oddly I couldn't find a definition of an interval over which anticipation was measured for the eye movements. What was the averaging interval? In the same vein, the goal of the fitting procedure is unclear to me. Why not average eye movemnets during an time-window corresponding to a period before visual information kicks in?

Figure 5 is quite confusing. It is trying to show too much. It would be more important to highlight the main conclusions, such as the comparison to ground truth, and the lack of correlation between h in the bet and spem estimate.

Line 606 to 616 this seems more like a discussion point. Also I am not clear how one of the statements is tested, regarding the fact variability in the estimate scales with inferred probability.

Reviewer #2: The authors presented a white ring that could move leftward or rightward. In one of the experimental sessions, the participants used smooth pursuit eye movements to track the motion of the ring. In a second experimental session, the participants were asked to “bet” or guess the direction of motion by placing a cursor on a continuous rating scale. The anticipatory smooth eye movements of the participants were modeled using a forgetful agent and a Bayesian model agent. The quantitative analysis of the authors showed close agreement between their Bayesian model agent and the participants’ anticipatory smooth eye movements and ratings.

Some of the strengths of this manuscript is that they made reasonable assumptions for their models. I was able to follow the logic behind the formulas and assumptions that they made. Directly comparing and modeling unconscious and conscious motor anticipatory behavior is a relatively novel and an important contribution. However, there are a few parts of the manuscript that need to be clarified.

Figure 3 needs to be reworked or the data needs to be presented differently. The way it is presented in the manuscript is too complicated. It took me too long to figure out what all of the lines mean, and I’m still not sure that I understand what the figure is showing. I read what the manuscript says about this figure and it may make sense, but it is difficult to see it in graph. I’m not saying that they authors are wrong or right; I just can’t understand what’s going on in the figure. Perhaps the authors can move this figure to the supplemental material and just show a simplified version of this figure in the manuscript. What does the negative and positive velocities indicate? Direction of motion?

I also had trouble following what’s going on in Figure 5. Can this figure have another panel showing the anticipatory smooth eye movements (gains?) the authors get with their paradigm for each individual subject? Or maybe just have another figure showing if there are any sequential effects? What does the anticipatory smooth eye movements look like when there is no switch for a relatively large number of trials? Is there a relationship between participants who tended to be more sure/confident about their assumption of the upcoming direction of motion? Do these results agree or disagree with previous studies?

Another part of the manuscript that can be clearer is the paradigm of the experiment. Starting at line 863, the manuscript says, “the moving target, which was presented slightly offset from the fixation location and immediately started moving horizontally at a constant speed of 15°/s, either to the right or to the left for 1000 ms.” Can you clarify this? Is this a “step-ramp”? Or, is it that the target appears off center and then begins to move? I was wondering if some of the participants of the study could have noticed the slight offset and used it as a cue for the direction of motion. For example, white ring is slightly to the left of the fixation cross therefore it will move toward the right. I know of at least one study that has used location of a cross as a cue to direction of motion for anticipatory smooth eye movements. If this is the case, the authors may need to disentangle what is anticipatory smooth eye movements in response to past history or in response to a cue. Could this explain some of the individual differences that the authors found? I was also looking at how much bigger the error bar plots for the “Bet score” are in Figure 4 than the “velocity of eyes”. Do the authors have any idea why this could be? I also didn’t understand how “strength of aSPEM” was calculated. Why is the median being used (instead of the mean)?

The manuscript would also benefit from adding a paragraph briefly describing the progression/evolution of models of anticipatory smooth eye movements and how their model will fit (agree or disagree) with existing models.

Minor:

There are few places in the manuscript where the word “prove” is used and it shouldn’t be used. For example, lines 11, 21… please replace/rephrase.

Remove extra space in line 426

Line 630 has a typo: “exploration versus exploration”

**Have all data underlying the figures and results presented in the manuscript been provided?**

Reviewer #1: Yes

Reviewer #2: Yes

PLOS authors have the option to publish the peer review history of their article (what does this mean?). If published, this will include your full peer review and any attached files.

Reviewer #1: No

Reviewer #2: No

---

## [Decision Letter · Decision Letter 1]

27 Feb 2020

Dear Dr. Perrinet,

We are pleased to inform you that your manuscript 'Humans adapt their anticipatory eye movements to the volatility of visual motion properties' has been provisionally accepted for publication in PLOS Computational Biology.

Please make sure that you check and fix the final issue raised by reviewer #1 before submitting your production material (I decided to give the manuscript the 'provisional acceptance' status nonetheless, to speed up the process rather than going throguh another round of minor revisions).

Best regards,

Wolfgang Einhäuser

Deputy Editor

PLOS Computational Biology

Reviewer's Responses to Questions

**Comments to the Authors:**

Reviewer #1: The authors have done a really nice job in their revision.

Minor point: In Figure 4, shouldn't "Preal" in the inset be "Ptrue"?

Reviewer #2: The authors of the manuscript have answered all of my questions. Congratulation on a very nice paper!

**Have all data underlying the figures and results presented in the manuscript been provided?**

Reviewer #1: Yes

Reviewer #2: None

PLOS authors have the option to publish the peer review history of their article (what does this mean?). If published, this will include your full peer review and any attached files.

Reviewer #1: No

Reviewer #2: No

---

## [Editor Report · Acceptance letter]

2 Apr 2020

PCOMPBIOL-D-19-01612R1 

Humans adapt their anticipatory eye movements to the volatility of visual motion properties

Dear Dr Perrinet,

I am pleased to inform you that your manuscript has been formally accepted for publication in PLOS Computational Biology. Your manuscript is now with our production department and you will be notified of the publication date in due course.

With kind regards,

Matt Lyles
